# Short-Term Effect of Fly Ash from Biomass Combustion on Spring Rape Plants Growth, Nutrient, and Trace Elements Accumulation, and Soil Properties

**DOI:** 10.3390/ijerph20010455

**Published:** 2022-12-27

**Authors:** Małgorzata Szostek, Ewa Szpunar-Krok, Marta Jańczak-Pieniążek, Anna Ilek

**Affiliations:** 1Department of Soil Science, Environmental Chemistry and Hydrology, College of Natural Sciences, University of Rzeszów, Zelwerowicza 8b, 35-601 Rzeszów, Poland; 2Department of Plant Production, College of Natural Sciences, University of Rzeszów, Zelwerowicza 4, 35-601 Rzeszów, Poland; 3Department of Botany and Forest Habitats, Faculty of Forestry and Wood Technology, Poznań University of Life Sciences, Wojska Polskiego 71f, 60-625 Poznań, Poland

**Keywords:** fly ash from biomass combustion, soil properties, soil solution, trace elements accumulation, spring rape

## Abstract

The short-term impact of biomass combustion fly ashes (BAs) fertilization on the spring rape growth, essential and trace elements accumulation in seeds, and selected soil and soil solution properties were analyzed in a pot experiment study. The pot experiment was carried out in the growing season (April–August) during the year 2018. The effect of BAs on the dry matter content in spring rape plants and the relative content of chlorophyll in leaves (SPAD) was analyzed. In addition, the effect of BAs on the accumulation of essential and trace elements in the seeds of this plant was analyzed. The impact of BAs on the basic physicochemical properties of soils was also assessed. Additionally, the solubility of compounds contained in BAs was monitored on the basis of the analysis of the changes in the physicochemical properties of soil solution during the experiment period. The present study demonstrated a positive effect of BAs fertilization on plant growth and development and improvement of soil physicochemical properties. A change has been achieved in the soil reaction class from a slightly acidic (control, NPK) to neutral (D1-D6), with the highest increase in pH induced by the highest ash dose of 3 mg ha^−1^ (D6). It was shown that BAs contributed to a significant increase in the content of macroelements than trace elements in the analyzed soil. In turn, the accumulation of these elements in plant seeds exhibited an inverse relationship, which was mainly influenced by the soil pH and the content of N, Ca, Mg, K, and Na in the soil, as indicated by the correlation coefficients. The highest contents of Fe, Mn, Zn, Cu, Cr, and Ni were detected in the seeds of plants fertilized with BAs at a dose of 2.0 Mg ha^−1^ (D4), and their respective values were 263, 363, 107, 51, 1835, and 137% higher than in the control. The Ca, Mg, S, and Na compounds introduced with BAs exhibited high solubility, as evidenced by the higher concentration of Ca^2+^, Mg^2+^, Na^+^, and SO_4_^2−^ ions in the soil solutions and the dynamic changes in pH and EC observed during the experiment. The lowest solubility after the application of BAs was exhibited by N and P. The conducted PCA analysis to a large extent explained the variability between the applied fertilization and the factors analyzed in the experiment. Despite the positive impact of ashes, attention should be paid to the potential risks associated with their use. The use of higher doses of BAs may result in excessive alkalization and salinity of soils and may enhance the accumulation of trace elements in plants. These aspects should therefore be closely monitored, especially in the case of a long-term application of these wastes, in order to avoid serious environmental problems.

## 1. Introduction

Due to the depletion of traditional non-renewable fossil fuels emitting large amounts of greenhouse gases, it is necessary to search for alternative environmentally friendly energy sources, with special emphasis to biomass [1,2]. Energy obtained from plant biomass combustion is increasingly being produced due to the necessity of transition to a low-emission economy [3]. The combustion of biomass and other energy carriers leads to the production of ashes at the annual level of 480–500 million tons. The management of ashes should be safe for the natural environment and for the health of humans and animals. According to the European Directive 2008/98/EC, ashes are considered as solid waste; hence, almost 70% of their amounts are stored in landfills [4]. This may cause serious damage to the natural environment, which may indirectly threaten human and animal health [2,5,6]. In addition, landfilling of ashes may not be possible in the future given the EU’s circular economy policy instruments [4]. Therefore, methods for rational management of these wastes should be developed. From the perspective of sustainable development, the most beneficial approach regarding the disposal of biomass ashes (BAs) is their return to the soil, and this solution is compatible with the circular economy idea too if proven to be economical and beneficial [7].

BAs are a solid residue of biomass combustion. It is a complex inorganic–organic mixture with poly component, heterogeneous, and variable composition containing intimately mixed solid, liquid, and gaseous phases with different origins [8]. BAs differ in their physical properties and chemical composition from ashes produced by burning traditional fossil fuels. Due to the higher content of potential plant nutrients than in bottom ashes, fly ashes are more suitable for soil fertilization and agricultural applications [1,5,8].

The chemical composition of BAs includes major elements (>1%), minor elements (1–0.1%), and trace elements (<1%) [8]. Fly ashes from biomass combustion and biomass itself may comprise as many as 79 different elements. Ashes from plant biomass combustion contain large amounts of macroelements and microelements necessary for plant growth and development, e.g., phosphorus, potassium, calcium, sulfur, magnesium, and others [1,2,7]. In turn, the presence of various types of harmful substances, such as heavy metals, limits the possibility of using BAs for the fertilization of plants [1,2,7]. Therefore, the chemical composition of ashes should be analyzed every time before using these wastes [7,9].

BAs were reported to affect the physical properties of soil due to their high capacity to absorb water and can change water relations in soil, improve soil porosity, and facilitate penetration of the soil by plant roots [9,10]. They also have an impact on physicochemical properties of soil, as they increase its pH thus producing a deacidification effect, increase the degree of saturation of the sorption complex with alkaline cations, and have diverse effects on the properties of organic matter, as widely documented in the literature [11,12,13]. Additionally, BAs directly influence the abundance of plant-available nutrients in the soil, i.e., macro- and microelements, in particular K, P, Ca, Mg, Mn, Z, and Fe [11,12,13,14,15,16]. The biological properties of soils can be improved with the use of BAs, as the ashes exert an impact on the microbiological structure of soil and enhance the activity of soil microorganisms [10,11,14,17].

Given the diversity of their chemical composition, the impact of fly ashes from biomass combustion on the soil should be closely monitored, mainly due to their potential to cause extreme changes in soil reaction and salinity or excessive accumulation of trace elements. Such long-term changes may bring many negative effects resulting in loss of soil fertility and productivity. Therefore, the main objective of the study was to assess the short-term impact of BA fertilization on the properties of soils and soil solutions. Additionally, the suitability of BAs for fertilization of spring rape plants was assessed and their effect on the accumulation of macroelements and trace elements in this crop was compared.

## 2. Materials and Methods

### 2.1. Pot Experiment Design

The pot experiment was carried out in the growing season (April–August) during the year 2018. It was carried out in a completely randomized design on brown soil with a granulometric composition of loamy dust (42% sand, 53% silt, and 5% clay) [18]. Approximately 30 kg of soil material was sieved through sieves with a mesh size of 4 mm and placed in plastic planting pots with a volume of 30 l (height 30 cm, diameter 37 cm). The physicochemical properties of the soil material used in the pot experiment are presented in Table 1.

The soil material was prepared to achieve 55% of the maximum water capacity and left for 14 days. BAs, were applied (2 April 2018) after this time. Fly ash collected from an electrostatic precipitator of a fluidized bed furnace was used in the experiment. The ash was produced through the combustion of forest biomass (approx. 70%) and agricultural biomass (approx. 30%). The forest biomass comprised deciduous and coniferous trees (50/50), whereas cereal straw, sunflower, husk, and willow were the components of the agricultural biomass. The characteristics of the ash used in the experiment are presented below in Table 2.

The variants of the experiment differed in the dose of potassium fertilization; this component accounted for the largest proportion of the ash used in the experiment which is related to biomass combustion technology [20]. The following doses of biomass combustion ashes were applied in the experiment: 0.5 (D1), 1.0 (D2), 1.5 (D3), 2.0 (D4), 2.5 (D5), and 3.0 (D6) Mg ha^−1^, which corresponded to the following doses of potassium applied as K_2_O: 100, 200, 300, 400, 500, and 600 kg ha^−1^, respectively. The pots from the different D1-D6 variants of the experiment were supplemented with 7, 14, 21, 28, 35, and 42 g of ash per pot^−1^, respectively. The amounts of nutrients introduced with the ashes to the individual pots are shown in Table 3. In all variants of the experiment, constant N and P fertilization with monoammonium phosphate NH_4_H_2_PO_4_ (12% N, 26.6% P) was applied in a total dose of 150 kg ha^−1^ before sowing. The effect of the biomass combustion ashes was compared with the control, i.e., potassium non-fertilized soil, and with soil receiving traditional potassium fertilization NPK. In this variant, the soil was fertilized with a dose of 175 kg ha^−1^ of 60% potassium salt. This dose of potassium results from the nutritional needs of spring rape plants.

After the application of individual fertilizers in the experimental variants, the soil was left for 14 days. Next, spring rape seeds were sown at a rate of 75 plants/m^2^ (10 seeds/pot) on April 17. The pots were then placed in a growth chamber (Model GC-300/1000, JEIO Tech Co., Ltd., Seoul, Republic of Korea) at 22 ± 2 °C, humidity 60 ± 3%RH, a photoperiod of 16/8 (L/D) h, and an ap-prox. 300 μE m^−2^ s^−1^ light intensity maximum. The pot positions in the growth chamber were randomized every week. In the stage of 2 leaves (BBCH11), the plants were thinned, leaving 5 rapeseed plants in each pot. The plants were maintained until seed maturity (BBCH89), which was reached after approx. 130 days from sowing.

The experiment were carried out in one series with three repetitions in each of the experimental variants.. Throughout the experiment, the soil moisture content corresponds to 55% of the maximum water capacity due to the high need for water for spring rape plants.

### 2.2. Analysis of Plant Samples

#### 2.2.1. Relative Chlorophyll Content (CCl)

The measurements were performed using a Chlorophyll Content Meter CCM-200plus (Opti-Sciences, Hudson, NH, USA). The relative chlorophyll content (SPAD) was measured on fully expanded wheat and rape leaves. Five leaves per pot were analyzed during every month of the experiment.

#### 2.2.2. Analysis of the Mineral Composition of Seeds and Aboveground Parts of Rape Plants

At the end of the experiment, the plants were carefully removed from the pots, and the aboveground parts were separated from the underground parts. In the aboveground parts, the dry matter was determined with the dryer-weight method by drying the plants at 45 °C. The dried seeds were homogenized in a laboratory mill and mineralized in 60% HNO_3_. The contents of basic macroelements, microelements, and trace elements (Ca, Mg, K, Na, Fe, Mn, Zn, Cu, Cr, Ni, Pb, and Cd) were determined in the mineralizes with the atomic absorption spectrometry technique using the HITACHI Z-2000 apparatus (Tokyo, Japan). The phosphorus content in the seeds was determined using the vanadate-molybdate method [20]. Nitrogen and sulfur contents in the aboveground and underground parts of the plants were determined with the dry combustion method using a Vario El Cube elemental analyzer (Elementar Analysensysteme GmbH, Langenselbold, Germany).

### 2.3. Soil Properties Analysis

Soil samples were collected after plant harvesting at the end of the experiment. The ash and soil samples were air-dried and sieved through a 2 mm diameter mesh (ASTM standards). Further, the samples were homogenized and stored in plastic bags until further analysis. The ash and soil samples were analyzed to determine the values of some parameters. The pH value was determined in a 1:2.5 substrate-water suspension using a 4221 pH-meter (Hanna Instruments, Nusfalau, Romania). Electrical conductivity (EC) was analyzed in a 1:5 substrate-water suspension with a HI 2316 EC-meter from Hanna Instruments (Nusfalau, Romania). The total nitrogen content (N) was determined with the Kjeldahl method [21]. The contents of macro- and trace elements (Ca, Mg, K, Na, Fe, Mn, Zn, Cu, Cr, Ni, Pb, and Cd) in the soil samples were determined with the absorption spectrometric method using a polarized Zeeman atomic absorption spectrophotometer Hitachi Z-2000 model (Tokyo, Japan) after mineralization of the soil samples in hydrochloric acid. The total phosphorus content in soil was determined using the vanadate-molybdate method [22]. Additionally, the content of trace elements soluble in water and 1 M HCl was determined in BA samples with the absorption spectrometric method as described above. The extraction procedure for soluble trace elements was applied to a mixture of BAs and the soil solution at the ratio (*w*/*v*) of 1:10. The samples were placed on a shaker and shaken for 24 h. After this time, the samples were filtered and the content of trace elements in the resulting solution was determined.

### 2.4. Soil Solution Samples Analysis

The soil solutions were collected on 10, 40, 70, 100, and 130 days after sowing the spring rape seeds. Once a week, in order to obtain samples of soil solutions, 200–300 mL of water more than necessary to maintain 55% of the maximum water capacity, was poured into each pot. The first determination was carried out 10 days after sowing the seeds. Each week of the following months, the solutions were collected in a plastic bottle and stored in a refrigerator at 4 °C until further analysis.

In the soil solutions, pH was determined with the potentiometric method using a 4221 pH-meter (Hanna Instruments, Nusfalau, Romania) and electrolytic conductivity (EC), which is a measure of salinity, was measured with the conductometric method using a HI 2316 EC-meter from Hanna Instruments (Nusfalau, Romania). The filtrates were analyzed for the content of inorganic ions: cations (Ca^2+^, Mg^2+^, K^+^, Na^+^, NH_4_^+^) and anions (Cl^−^, NO_3_^−^, PO_4_^3−^, SO_4_^2+^) with the ion chromatography method using the DIONEX 5000+ apparatus.

### 2.5. Statistical Analysis

All statistical analyses were performed using STATISTICA 13.3 software (StatSoft, Tulsa, OK, USA). To show the existence of uniform groups of objects (α = 0.05), the Tukey multiple comparison HSD test was performed following a two-dimensional analysis of variance (ANOVA). In order to estimate the sources of variation, Pearson’s correlation (r) and principal component analysis (PCA) were additionally performed.

## 3. Results

### 3.1. Physicochemical Properties of BAs

The BAs used in the experiment were characterized by higher alkalinity and salinity (pH = 12.83, EC = 8.81 mS cm^−1^). The macroelements required for plant growth and development contained in the ashes exhibited the following order of decreasing values: K > Ca > Mg > P>S > Na > N; their mean amounts were estimated at 165,617, 145,081, 13,512, 9244, 4700, 1452, and 10 mg kg^−1^. The total content of trace elements (Fe, Mn, Zn, Cu, Cr, Ni, Cd, and Pb) was on average 4351, 1490, 423, 536, 20.3, 48.4, 130, and 2.68 mg kg^−1^ (Table 2). The trace elements contained in the ashes were characterized by very low solubility in water, and their amount in relation to their total content did not exceed 0.5%. In turn, the amount of trace elements soluble in 1 M HCl varied and accounted for 28, 43, 93, 29, 78, 68, 26, and 93% of the total content of Fe, Mn, Zn, Cu, Ni, Cr, Pb, and Cd, respectively.

### 3.2. Effect of BAs on Plant Growth and Development and Grain Mineral Composition

The mean dry matter (d.m.) content in the spring rape plants, ranging from 8.98 to 20.79 g, was significantly correlated with the applied fertilization (Figure 1). The lowest values were noted in the control object, whereas variant D3 exhibited the highest values. The effect of potassium introduced with BAs on plant growth was similar to that of the NPK mineral fertilization, as evidenced by the absence of significant differences between these experimental objects. It was also found that the introduction of both too low (D1) and too high (D5 and D6) doses of ash resulted in a slightly weaker increase in the dry matter of the plants than in the other experimental objects (D2-D4), where the best results were achieved. In comparison with these objects, the mean dry matter content in the control had a two-fold lower value (Figure 1).

Similar relationships were observed in the case of the relative content of chlorophyll (SPAD) in the rapeseed leaves (Figure 2). The mean SPAD values (on average: 17.12 relative units) in the control were significantly lower than in the other experimental objects. No statistically significant differences were found between the object receiving traditional NPK fertilization and the BA-fertilized variant, where the mean SPAD values ranged from 20.27 to 23.32 (Figure 2). As in the case of dry matter, better results were achieved in objects D2-D4 than in variants fertilized with the lowest and highest doses.

The mean content of the analyzed elements (N, P, K, Mg, Ca, Na, and S) in the spring rape seeds was in the range of 29.0–31.5, 3.78–7.72, 8.15–12.14, 1.20–3.01, 4.13–5.09, 0.12–0.22, and 1.55–2.32, respectively (Table 3). In comparison with the control, the mean values of the parameter were generally higher in plants fertilized with traditional NPK fertilizers and BAs. In the group of the analyzed elements, only the K content increased linearly together with the increasing doses of BAs. Moreover, the mean levels of this element in the BA-fertilized plants were significantly higher than in the control and the NPK object, except for object D1. Similar relationships were found in the case of Ca. The lowest mean P content in the spring rape seeds was determined in the D1 and D2 treatments, and these values were significantly lower than in both the control and the NPK fertilization variant. The lowest mean Mg content in the spring rape seeds was noted in the control. It increased significantly as a result of the application of both NPK and BAs. The comparison of the NPK-fertilized objects with those fertilized with BAs showed significantly higher content of this element in objects D1-D4 than in the NPK fertilization variant. The content of this element after the application of the highest doses of BAs (D5 and D6) was similar to that in the NPK-fertilized objects. The application of BAs, especially in the highest doses (D4-D6), contributed to a significant increase in the mean Na content in the spring rape seeds. In turn, an opposite trend was found in the case of the mean S content: the application of the highest doses of BAs (D4-D6) resulted in a significantly lower mean level of this element than in the other experimental objects.

The analyzed objects did not differ significantly in the mean content of N in the spring rape seeds. The mean content of this element was significantly lower only in object D3, in comparison with the control, NPK, D5, and D6 variants (Table 3).

Compared to the control, the traditional NPK fertilization and the BAs treatment significantly increased the mean content of trace elements in the spring rape seeds (Table 4). The highest mean content in the spring rape seeds was determined in the case of Fe, with values ranging from 22.6 (control) to 82.1 (D4) mg kg^−1^. The mean content of this element in the NPK and D1 variants was similar to that in the control. The D2-D4 doses contributed to a significant increase in the content of this element in the rape seeds, and the values achieved in the D4 object were nearly 4-fold higher than in the control, NPK, and D1.

In comparison with the control and NPK variants, the addition of BAs significantly increased the mean Mn content in the spring rape seeds. The mean content of this element varied in a wide range from 6.1 (control) to 28.3 (D4) mg kg^−1^. The fertilization with BAs resulted in an approximately four-fold increase in the mean content of this element, in comparison with the control and NPK plants (Table 4). Similar relationships were exhibited by the Cr content, which increased significantly after the application of BAs. The highest content of this element (on average: 3.87 mg kg^−1^) was detected after the application of the D4 dose; it was nearly 20-fold higher than in the control and the NPK object. In comparison with the control, the fertilization had a significant effect on the mean Zn content in the seeds in the traditional NPK fertilization and BAs treatment variants. The comparison of the NPK and BAs fertilization variants revealed significantly higher contents of this element only in plants fertilized with the lowest doses (D1 and D2). Similar relationships were determined in the case of Cu. In comparison with the NPK variant, its mean content (2.97 mg kg^−1^) was significantly higher only in object D1. Similarly, the mean content of Ni in the rape seeds increased after the application of BAs. As in the case of Fe, Mn, Zn, and Cr, the mean content of this element increased up to the D4 dose and decreased in the D5 and D6 variants. In contrast to the elements discussed above, the mean Pb content in the seeds exhibited a slightly different tendency, as the lowest levels of this element were observed in object D4 and the highest content was determined at the highest doses of ashes (D5 and D6). These values were significantly higher than in the other experimental objects. Similar relationships were exhibited by Cd: the lowest levels of this element were detected in plants from the traditional NPK fertilization variant (Table 4).

To estimate the sources of variability, the relationship between the plant parameters and the applied fertilization was assessed using the principal component analysis (PCA) (Figure 3). The dry matter of the spring rape plants, SPAD, and the content of macro- and microelements in the seeds were taken into account in the analysis (Figure 1 and Figure 2, Table 3 and Table 4). Three main components with a greater variability load than the individual variables were distinguished among the analyzed parameters. The eigenvalues of the selected components were greater than 1. The first and second principal components explained 43.93% and 20.77% of the variance, respectively. The first three principal components explained a total of 78.74% of the variance in the raw data. The content of Cr, Mg, Mn, Fe, and K, the dry mass of the aboveground parts, and the relative chlorophyll content constituted the first main component. The second component comprised mainly the content of S, Zn, K, Na, Ni, and Pb in the seeds and the relative chlorophyll content. The content of Cd, Pb, Fe, P, N, and Cu in the seeds had a major impact on the formation of the third main component. Over 90% of the variance in the Cd, Cr, K, Mg, and Zn levels was explained by the three principal components. The weakest explanation of the variance was provided by the three principal components in the case of the N content (17%) (Figure 3).

### 3.3. Effect of BAs on Soil Properties

The application of BAs resulted in significant changes in soil pH (Table 5). In comparison with both the control and the NPK object, a significant increase in pH was observed upon the treatment with the lowest dose (D1). As a result of the fertilization with the biomass ash, the pH class changed from slightly acidic (control, NPK) to neutral (D1-D6). The average pH value after the application of the highest dose (D6) was 7.29, which was 0.81 and 0.70 higher than in the control and the NPK variant, respectively. The ashes from biomass combustion also induced changes in the average EC values, which ranged from 0.10 (control) to 0.49 (D1) mS cm^−1^. With the exception of variant D6, the average EC values in all the other ash-fertilized objects (D2-D5) were significantly higher than in the control and the NPK object. According to the agronomic classification of soil salinity based on the EC value [23], the soil from all the experimental treatments represented the non-saline soil class. Despite the increase in the average EC values, the BAs fertilization did not change the salinity class (Table 5).

The treatments with both the traditional NPK fertilizers and the biomass combustion ashes generally led to an increase in the content of N and P in the soil, compared to the control, and the average contents of N and P were in the range of 1927–2302 and 979–2326 mg kg^−1^, respectively. The present data indicate that traditional NPK fertilization contributed to a significant increase in the average content of N and P as well. In comparison with this object, significantly higher N values were recorded in the soils fertilized with the D4 and D6 BA doses, and significantly higher P values were noted only in the object fertilized with the highest dose of ashes (D6); the contents of the elements in the other variants were lower or comparable (Table 5).

A different tendency was observed in the case of the other elements, i.e., K, Mg, Ca, and Na, whose average content was in the range of 1110–1580, 1247–1491, 1481–1865, and 136–171 mg kg^−1^, respectively. The lowest levels of these elements were determined in the NPK fertilized soil. The changes in their content in the soil were largely influenced by the BA fertilization, which significantly increased their amount in the soil and the increase was correlated with the applied dose (Table 5).

The application of BAs in all the analyzed doses significantly increased the average content of Fe, Mn, and Ni in the soil in comparison with both the control and the traditional NPK fertilization variant (Table 6). Moreover, the levels of Zn, Cu, Cr, and Pb in the NPK object were lower than in the control, but no statistical significance of these differences was confirmed. The concentration of Fe, Mn, Zn, Cu, Ni, Cr, Pb, and Cd in the soil was in the range of 1983–2916, 120–204, 40.4–49.5, 4.54–6.27, 14.6–19.0, 26.3–31.9, 14.3–16.7, and 0.22–0.34 mg kg^−1^, respectively. The highest concentrations of Fe, Mn, Zn, Ni, Pb, and Cd were determined in the soil fertilized with the D4 dose. In turn, the highest mean content of Cr was noted in object D2, whereas the objects fertilized with the highest ash doses (D5 and D6) exhibited the lowest level of this element. An increase correlated with the increasing doses of BAs was observed only in the case of Cu. The experimental variants were found to have diverse effects on the content of Pb and Cd in the soil. In comparison with the control and the NPK object, the content of Pb and Cd significantly increased in the soil in the D4 and D6 variants; additionally, an increased level of Cd was noted in the object fertilized with dose D1. No significant changes in the content of these elements were shown by the other variants of the experiment (Table 6).

The principal component analysis (PCA) was carried out to estimate the sources of variability between the soil properties and the fertilization variants (Figure 4). Soil properties such as pH, EC, and the content of the macro- and trace elements were analyzed (Table 5 and Table 6). Three main components explaining the highest percentage of variability were distinguished among the sixteen analyzed parameters. The first principal component (F1) explained 50.96% of the variance, whereas the second and third principal components (F2 and F3) explained 16.13% and 12.57% of the variance, respectively. In total, 79.6% of the variance was explained by the principal components. The largest amount variance (67.08%) was explained by the first two principal components. The first principal component explained approximately 80% of the variance in the content of Ca, Mg, and Zn in the soil, but the variability of EC and the Cr content in the soil was explained more weakly. The second principal component increased the level of explanation of the variance to over 90% in the case of Mg and K and over 80% in the case of pH, Na, Ca, Zn, and Cd. The third principal component increased the proportion of explanation of the EC variance up to 90%. This component also explained over 90% of the variance of pH, Zn, and Ni. The weakest explanation by the three main components was noted in the case of the content of Cr in the soil (30%).

### 3.4. Relationship between Plants and Soil Parameters

Soil pH and the content of N, P, and Ni were positively correlated with the dry matter content in the plants (Figure 5). The correlation coefficients indicated that the soil pH value and the N content had the greatest impact on the levels of almost all elements determined in the spring rape seeds. The increasing content of Na in the soil was correlated with an increase in the concentration of Ca, Cd, Cr, Cu, Fe, Mg, Mn, Na, and Zn in the rape seeds. In turn, the increase in the concentration of K, Ca, and Mg had a significant effect on the accumulation of Ca, Cd, Cr, Cu, Fe, K, Mn, Na, and Pb in the seeds. The increase in the Ca content in the soil induced by the introduction of the increasing BA doses resulted in a decrease in the S content and an increase in the P content in the seeds; the decrease in the S content was additionally found to be influenced by Fe. It was also shown that the increase in the Cu content in the soil contributed to a higher concentration of Ca, Cd, Cr, Fe, K, Mg, and Mn in the seeds. In turn, the increasing concentration of Mn and Ni in the soil had a significant impact on the content of Cr, Cu, Fe, Mg, Mn, Ni, Pb, and P in the seeds. Moreover, the Cd contained in the soil increased the content of only Cu and Ni in the spring rape seeds (Figure 5).

The values in the square lattices represent the magnitude of the R value of correlation analysis displayed by color difference.

### 3.5. Effect of BAs on the Physicochemical Properties of Soil Solution Samples

The pH values of the analyzed soil solutions on days 10 and 40 after sowing the seeds ranged from 6.62–7.08 and 6.71–7.50, respectively, with the highest values in the control and NPK variants, but declined with the increasing ash doses. The analysis of samples on days 70, 100, and 130 from sowing the seeds showed that mean pH values were in the range of 6.45–7.76, 6.63–8.20, and 6.70–7.58, respectively, with the lowest values determined in the control, but increased after the application of BAs (Figure 6A).

The average EC values in the soil solutions in all the experimental objects exhibited at a similar level until 70 days after sowing the seeds. After 100 days, the EC values increased dynamically and reached the highest values in objects fertilized with the highest doses of BAs (D5 and D6). On day 130 after sowing the seeds, the EC values slightly decreased but were higher than in the initial stage of the study (Figure 6B).

The fertilization with the biomass combustion ash increased the content of Ca^2+^, Mg^2+^, K^+^, and Na^+^ in the soil solutions, in comparison with the control and NPK variants (Figure 7A–D). The content of these components changed dynamically during the experiment and, in the final stage, fluctuated in the objects fertilized with the different doses of BAs in the following ranges: 101.5–154.9, 8.2–12.9, 2.52–4.28, and 36.3–48.2 mg L^−1^. For comparison, the content of these components in the control and NPK treatment in the final stage of the experiment was 19.4 and 21.5, 1.5 and 1.9, 0.32 and 0.09, and 9.4 and 10.4 mg L^−1^, respectively. The most dynamic changes in the content of these components in the soil solutions were observed between days 40 and 100 after sowing the seeds. It can also be noticed that the content of the above-mentioned ions in the soil solution was declining successively at the end of the experiment. The most diverse changes were observed in the concentration of NH_4_^+^ ions in all the experimental objects. Their elevated concentrations were usually found between days 40 and 100 after sowing the seeds (Figure 7E).

The application of the ashes from biomass combustion contributed to the leaching of Cl^−^ and SO_4_^2−^ ions, which were present in large amounts in the analyzed soil solutions (Figure 8A,D). The highest content of Cl^−^ ions in objects D1–D6 was determined at the beginning of the experiment, but it decreased significantly between days 40 and 100 after sowing the seeds. After this time, the concentration of Cl^−^ in the soil solution increased dynamically, and the average values recorded in the final stage of the experiment ranged from 65.3 (D2) to 168.3 (D3) mg L^−1^ (Figure 8A). A similar dynamics of changes was observed in the concentration of SO_4_^2−^ ions, whose average concentration at the final stage of the experiment in the BAs-fertilized objects was in the range of 301.4–544.4 mg L^−1^. These values were significantly higher than the mean SO_4_^2−^ concentrations of 20.8 and 9.2 mg L^−1^ in the control and NPK variants, respectively (Figure 8D). Diverse changes were observed in the concentration of NO_3_^−^ and PO_4_^3−^ ions (Figure 8B,C). The highest concentration of NO_3_^−^ was determined in the initial stage in the control and NPK variants and between days 70 and 130 after sowing the seeds in object D4 (Figure 8B).

Regardless of the experiment variant, the lowest leaching rate was noted in the case of PO_4_^3−^ ions, as evidenced by their varied dynamics observed during the experiment (Figure 8C). No PO_4_^3−^ ions were detected in the soil solutions at most of the analyzed time points. On day 10 after sowing the seeds, these ions were released only in the control and NPK variants and their content was determined at 0.38 and 0.22 mg L^−1^, respectively. On day 100 after sowing of the seeds, PO_4_^3−^ ion leaching was observed exclusively in objects D3–D6, and their content ranged from 0.16 to 0.41 mg L^−1^ (Figure 8C).

The principal component analysis (PCA) was performed to estimate the sources of variability between the properties of the soil solutions (pH, EC, content of inorganic ions) and the applied fertilization (Figure 9). Three main components explaining the largest percentage of the variance were distinguished among the eleven analyzed parameters. The first principal component (F1) explained 37.08% of the variance, while the second and third principal components (F2 and F3) explained 17.27% and 11.97% of the variance, respectively. In total, 66.32% of the variance was explained by the three principal components.

The greatest proportion of the variance was explained by the first two principal components–54.35%. The first principal component explained over 70% of the variance for Ca^2+^, Cl^−^, and SO_4_^2−^ ions contained in the solution. The inclusion of the second principal component helped to explain over 80% of the variance of these ions and Na^+^ ions contained in the soil solution. The third principal component explained a greater proportion of the pH and EC variance. The principal components explained the lowest proportion of variability of NO_3_^−^, PO_4_^3−^, and NH_4_^+^ ions contained in the soil solution. The percentage of the variance explained for these variables was 22, 33, and 46%, respectively (Figure 9).

### 3.6. Relationship between Soil Parameters and Soil Solution Properties

The soil pH value was found to exert a significant impact on changes in the properties of the soil solutions. A significant negative correlation was found between the soil pH and soil solution pH and the content of NH_4_^+^, NO_3_^−^, and PO_4_^3−^ ions. The increase in soil pH had a direct impact on the EC of the soil solutions and the content of Ca^2+^, Mg^2+^, Cl^−^, and SO_4_^2−^ ions. In turn, the increase in the EC value after the application of BAs resulted in an increase in the concentration of Na^+^ ions in the soil solution. Additionally, the increase in the content of macroelements in the soil had a significant effect on the concentrations of almost all inorganic ions and changed the pH and EC values in the soil solutions. The absence of significant correlations between the content of P in the soil and the content of PO_4_^3−^ ions in the soil solution may indicate relatively low solubility of P contained in the ash. In the group of the trace elements contained in the soil, mainly Cu as well as Zn and Ni were significantly positively or negatively correlated with the content of some inorganic ions in the soil solutions and with their pH and EC values (Figure 10).

The values in the square lattices represent the magnitude of the R value of correlation analysis displayed by color difference.

## 4. Discussion

Ash from biomass combustion contains substantial amounts of macro- and micronutrients; hence, it can be used for fertilization of different crop species (e.g., spring wheat, maize, potatoes, legume) [15,24,25,26]. The dose of ash used in fertilization schemes for various plant species is determined primarily through analysis of the content of phosphorus and potassium. BAs may contain over 40% of calcium compounds, up to 36% of potassium compounds, and 7% of phosphorus compounds [17]. It has also been indicated that fly ashes are more suitable for fertilization than bottom ashes, as the former contain much larger contents of macro- and microelements, although they may additionally contain larger amounts of impurities. The positive impact of biomass combustion ashes on the growth and development of various plant species has been widely documented in the literature and was also confirmed in the present study [12,15,16,25,26].

The biomass ashes used in the experiment contained large amounts of macro- and microelements (except N), and their application, starting from the dose of 1 Mg ha^−1^ (D1), resulted in more efficient plant growth and development, in comparison with the control variant. It was also found that the fly ash-induced increase in the dry matter content of plants, the SPAD value, and the accumulation of basic macronutrients in the spring rape seeds was comparable to that observed in the traditional NPK fertilization variant. This indicates that biomass ashes can be used successfully as a valuable fertilizer in spring rape plant cultivation. Nevertheless, in the case of dry matter and SPAD, better results were achieved in objects D2-D4 than in variants fertilized with the lowest and highest doses (Figure 1 and Figure 2). The use of ashes, especially in higher doses, does not significantly improve the SPAD value and dry matter content [24]. This could be attributed to the mineral composition of the ash, it having high concentrations of important elements but virtually no nitrogen. Furthermore, wood ash application may strongly influence the soil texture, aeration, and water holding capacity, consequently having an impact on the root growth dynamics leading to a range of possible effects on plant growth [24,27].

Despite the similarities in the effect of the traditional NPK and BAs on plant growth and development, these fertilizers exerted different effects on the analyzed soil properties. BAs contain high concentrations of soluble alkali metal salts, hydroxides, oxides, and carbonates, especially compounds with Ca, Mg and K, which have an impact on the pH and EC values of these fertilizers [5]. Most BAs have a pH value ranging from 9.4 to 13.5, and the ash used in the experiment had similar values (Table 2). BAs positively influence a number of soil properties, mainly by increasing the pH value which has been widely documented in the literature [12,14,28]. It has been emphasized that BAs can be a substitute for calcium fertilizers used for deacidification of both agricultural and forest soils [29]. BAs present in the soil react much faster than lime and contribute to a stronger but relatively short-term increase in pH [29]. The application of even the lowest dose of BAs in the present study resulted in an increase in the mean soil pH values. The soil reaction changed under the impact of the ashes from slightly acidic (control, NPK) to neutral (D1-D6), with the highest increase in pH achieved after the application of the highest ash dose of 3 Mg ha^−1^ (D6). It has been indicated that BAs have the most pronounced effect on the pH value in soils with higher acidity and low organic matter content [28]. As reported by Hansen et al. (2017), changes in pH in the topsoil exhibit the greatest dynamics during the first 50 days after BAs application. Noteworthy, the use of BAs may cause extreme pH fluctuations, especially in the topsoil layer (pH = 11.0–4.4). Therefore, this should be taken into account in assessment of the impact of fly ashes on pH changes and soil properties [29].

It is assumed that changes in the chemism of soil solutions in BAs-fertilized soils are good indicators of the solubility, mobility, and availability of plant nutrients as well as soil fertility [30,31,32]. The application of BAs may induce a rapid increase in the soil solution pH value, which persists for approximately two months after the fertilization [32]. The present study showed the opposite tendency, as the most pronounced changes in the pH of the soil solution after the application of BAs were noted only at approximately 70 days after sowing the seeds (Figure 4). This may be related to the characteristics of the ashes used in the experiment. They were derived from combustion of agricultural and forest biomass, and the compounds contained therein may have been less soluble. Changes in the pH of the soil solution as a result of the use of various waste materials, including ashes, may also be associated with changes in weather conditions, temperature, microbiological activity, as well as interactions that occur between individual compounds. Changes in the chemistry of the soil solution are also closely related to the metabolic processes of plants, mainly by substances secreted by the rhizosphere zone. It is indicated that changes in the pH of soil solutions may be associated with a decrease in the concentration of NO_3_^−^ ions, which are taken up by the roots of developing plants [32]. Some studies also indicate that the pH of the soil solution is more dependent on the type of soil and the species of a cultivated plant species, than the type of ash, used [32].

As indicated by Gómez-Rey and Coutinho (2012), fertilization of soil with ashes from biomass combustion, especially applied as a loose material, enhances N and P leaching, especially in the first month after application, which was not confirmed in the present study [33]. In the initial stage of the experiment, the average concentration of NO_3_^−^ ions was much higher in the control and NPK variants, which may have been related to the release of nitrogen compounds introduced with the mineral fertilizer. The lower content of these ions in the soil solution may indicate increased uptake thereof by plant roots on the one hand, as the plants exhibited more intensive growth after the application of the ashes [31]. On the other hand, they may indicate reduced leaching of nitrogen compounds from the soil due to the introduction of substances that block this process but do not reduce nitrogen assimilability. After application of ashes, Bielinska et al. (2009) reported extremely high content of N-NO_3_^−^ in soil, especially in a cereal cultivation variant. The authors explain this phenomenon by transformations of nitrogen, in particular by nitrification, which proceeds most intensively in aerobic conditions and in soils with a slightly acidic to neutral pH value and with a large amount of phosphorus [34]. Similar findings concerning the concentration of NH_4_^+^ in the soil solution were obtained in the present study. In this case, the mean concentration of NH_4_^+^ in the soil solution at the beginning of the experiment was many times higher in the control and NPK variants. It can be concluded that the compounds contained in the ashes inhibited leaching of nitrogen compounds into the soil solution. Despite the low content of N in the ashes, the limitation of its solubility could also contribute to the higher content of N in the soil, compared to the control (Table 5) [24]. As there were very small changes in the content of PO_4_^3−^ ions in the soil solutions, a relatively low rate of leaching of phosphorus introduced with BAs was assumed. Already on day 10 after sowing the seeds, PO_4_^3−^ ions were detected in the soil solution in the control and NPK variants but not in the BA-fertilized objects. The low rate of leaching of phosphorus applied with the ashes should have resulted in an increase in its content in the soil and higher accumulation in plants. However, the content of P was significantly higher only in the object fertilized with the highest BAs dose. The seeds of plants fertilized with the highest BAs doses exhibited higher accumulation of this element in comparison with the seeds from the other experimental objects. Some portion of phosphorus may have been accumulated by other parts of the plants, but this issue was not investigated in the present study. As shown by other researchers, P derived from BAs is very poorly soluble. In turn, a large portion of soluble P is probably immobilized in soil, which does not result in increased accumulation of P by plants. It has also been demonstrated that the uptake of phosphorus supplied with BAs is much less efficient than in the case of phosphorus provided by mineral fertilizers [28,29,30].

The BAs fertilization resulted in increased release of sulfur compounds into the soil solution, which resulted in high concentrations of SO_4_^2−^ ions in the solution. In the final stage of the experiment, their average content in the object fertilized with the highest ash dose (D6) was as much as 2615% higher than in the control. Noteworthy, the lowest average S content was detected in the plants fertilized with the highest doses of BAs, in comparison with the other experimental objects. This may additionally confirm the high solubility of S compounds contained in ashes, which results in sulfur leaching beyond the reach of the root system and thus lower assimilability of this element. Some of the components introduced with ash to soil are dissolved, which results in an increase in their concentration in the soil solution. It is believed that Ca and Mg are present in ashes mainly as less soluble secondary minerals [11]. Nevertheless, the results of the present study indicate relatively easy leaching of these elements, resulting in their substantial concentrations in the soil solution; this trend was found to persist throughout the experiment. The mean concentration of Ca^2+^ ions before plant harvesting in the object fertilized with the highest dose of ashes (D6) was 691% higher than in the control. Despite the high concentrations of Ca^2+^ in the soil solution throughout the experiment, at the end of the experiment, the content of Ca in the variants supplemented with the highest ash doses (D4–D6) was 18, 10, and 20% higher, respectively, than in the control. The application of BAs contributed to a significant increase in the content of Mg^2+^ ions in the soil solution, and this effect persisted throughout the experiment. As a result of the BAs fertilization, the average Mg content at the end of the experiment was significantly higher than in the control.

It has been evidenced that Na, K, and B are present in ash as easily soluble salts, which are released from ash particles within the first two years after application [11]. Large amounts of easily soluble salts of such elements as K and Na may be responsible for soil salinity, which should be considered in assessment of the impact of ash waste on the soil environment. If the availability of fertilizer ingredients exceeds plants’ needs, easily soluble compounds may leach out of reach of the root system, which is especially the case of potassium [11]. The BAs fertilizer used in the experiment contained nearly 17% of K. Despite the high solubility of K, no significant increase in the concentration of K^+^ ions in the soil solution was observed throughout the experiment. Due to its good solubility, K was accumulated in the spring rape seeds in a substantially larger amount than in the control and NPK treatments. Additionally, after the experiment, the average content of K in the soil treated with the highest dose of BAs was 32% higher in comparison with the control. Although the ash used in the experiment contained significantly lower content of Na than K (0.14%), a significant increase in the concentration of Na^+^ ions in the soil solution was observed during the study period. Before the end of the experiment, the average concentration of Na^+^ ions in the soil solution in objects D5 and D6 was 412 and 342% higher, respectively, compared to the control, and the average content of this element in the soil was approx. 70% higher. The present study also showed a higher concentration of Cl^−^ ions in the soil solution after the application of BAs, compared to the control and NPK variants. This higher concentration, detected throughout the experiment, was shown to be dependent on the ash doses. The Na^+^ and Cl^−^ ions contained in the soil solution contributed to a significant increase in the salinity of soil solutions. Dynamic changes in the content of Ca^2+^, Mg^2^+, K^+^, Na^+^, and Cl^−^ in the soil solution between 40 and 100 days after sowing the seeds also resulted in dynamic changes in the EC values of the soil solution (Figure 6B).

In turn, the increase in K and Na content in the analyzed soil increased the EC value, but the salinity class was unchanged by the ash fertilization. Due to the possibility of soil salinization induced by ashes, this parameter should be closely monitored.

In addition to the content of macroelements that are valuable for fertilization, BAs also contain large amounts of trace elements, some of which function as microelements while others are highly toxic to living organisms [31]. Fe as well as Mn, Cu, and Zn were the dominant microelements present in the ash used in the experiment (Table 2). Fe is a microelement present in BAs in the largest amounts, reaching up to 21 g kg^−1^ in ashes produced from wood biomass [29]. The use of BAs was found to increase the total content of Mn, Ni, Zn, and Cu in soil with no significant changes in the content of Cd and Pb [29]. Similar results were obtained in the present study; however, BAs was shown to increase the amounts of macroelements to a much greater extent than those of trace elements in the analyzed soil. Unlike in the case of macroelements, the accumulation of trace elements in the seeds was significantly higher in the ash-fertilized objects than in the control and NPK variants (Table 3 and Table 4). The highest accumulation of the Fe, Mn, Zn, Cu, Cr, and Ni elements was detected in plants fertilized with BAs at a dose of 2.0 Mg ha^−1^ (D4). Compared to the control, the average contents of these elements were higher by 263, 363, 107, 51, 1835, and 137%, respectively. It has been indicated that BAs may improve the availability of Cu and Fe in the soil as well as Pb [16]. The soil pH value was a probable cause of the lower accumulation of Fe, Mn, Zn, Cu, Cr, and Ni in seeds of plants fertilized with doses D5 and D6. It is well-known that higher pH values reduce the solubility and mobility of most trace elements [34,35,36,37]. Reducing the mobility of these elements as a results of the increase of soil pH could have caused their greater accumulation in the root zone, which would explain their lower concentration both in the soil and in the seeds [24]. The highest contents of Pb and Cd in the seeds were determined in the treatment with the highest doses of BAs. The Pb content in the seeds positively correlated with soil pH and K content and exhibited a negative correlation with the content of Cr in the soil. In turn, the level of Cd in the seeds was positively correlated with pH and the content of K, Ca, Mg, and Na in the soil (Figure 5). Literature data have demonstrated that rape plants have the ability to absorb large amounts of Zn, Cu, Pb, and Cd from the soil, which are accumulated in seeds and vegetative organs. Thus, they can be used to remediate soils contaminated with such elements as Cd, Cr, Cu, Ni, Pb, and Zn [36]. This was confirmed by studies [38,39] showing that the application of wood ashes resulted in hyperaccumulation of trace elements in rape plants in the following bioaccumulation order: Fe > Zn > Pb > Co > Cu > Cd > Ni.

The relatively high concentrations of Cd and Pb as well as Cr and Ni in BAs, their relative solubility, regardless of soil pH, and accumulation in spring rape seeds should be considered in assessment of the potential impact of biomass ashes on the environment [36].

## 5. Conclusions

The study revealed the following findings:The biomass ashes used in the experiment contained large amounts of macro- and micronutrients (except N), and starting from the dose of 1 Mg ha^−1^ (D1), the BAs application contributed to better plant growth and development compared to the control. Significant effectiveness of BAs in fertilization of spring rape plants, comparable to traditional NPK fertilization, was demonstrated. The utilization of nutrients contained in BAs is in line with the circular economy and fully justified.In addition to the supply of almost all essential nutrients to plants, short-term application of ashes positively changes the physicochemical and chemical properties of soils. Therefore, the application of this fertilizer in soil is consistent with the principles of sustainable development and may contribute to maintenance of the fertility and productivity of agricultural soils in many aspects.The application of BAs may increase the accumulation of alkaline substances in soil, especially those containing such elements as Ca, K, and Na, thereby causing excessive alkalization and salinization. There is also some risk that the BAs fertilization, especially in higher doses, may enhance the accumulation of trace elements in plants. These aspects should therefore be closely monitored, especially during a long-term use of these wastes, in order to avoid serious environmental problems.

## Figures and Tables

**Figure 1 ijerph-20-00455-f001:**
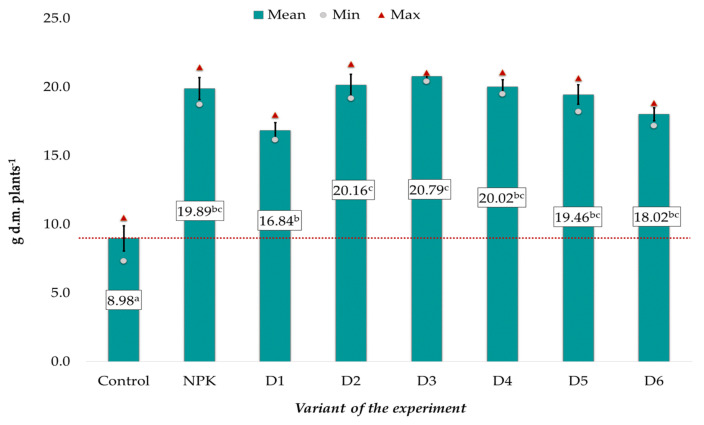
Dry matter of spring rape plants (d.m.) depending on the fertilization variant (mean ± SE). Mean values ± standard error. (a–c) Identical superscripts denote no significant (*p* < 0.05) differences between the experimental objects according to the post-hoc Tukey HSD test. The red line represents the value in the control.

**Figure 2 ijerph-20-00455-f002:**
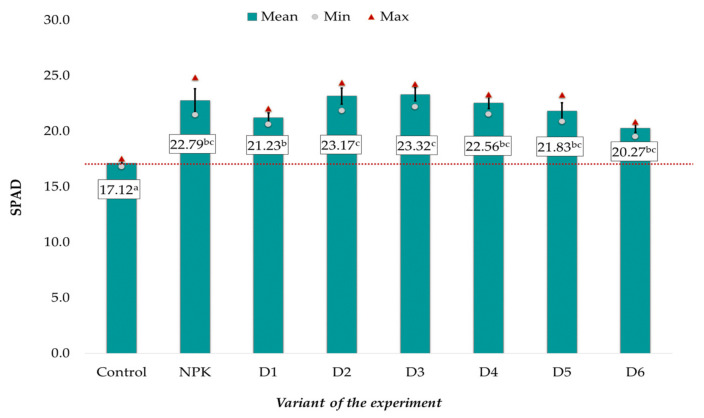
Relative chlorophyll content in spring rape leaves depending on the fertilization variant (mean ± SE). Mean values ± standard error. (a–c) Identical superscripts denote no significant (*p* < 0.05) differences between the experimental objects according to the post-hoc Tukey HSD test. The red line represents the value in the control.

**Figure 3 ijerph-20-00455-f003:**
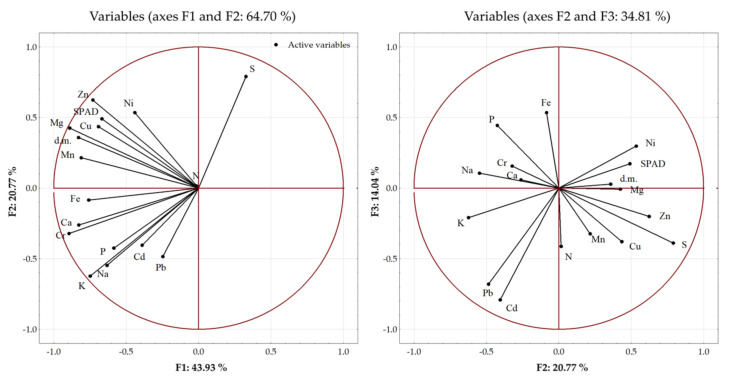
Principal component analysis (PCA) biplot of the distribution of the analyzed plant parameters.

**Figure 4 ijerph-20-00455-f004:**
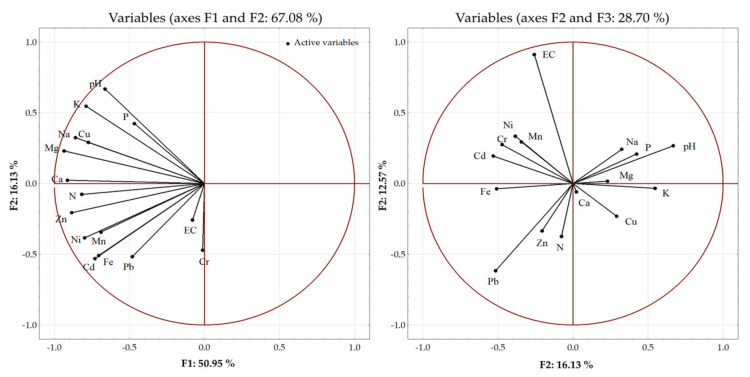
Principal component analysis (PCA) biplot of the distribution of the analyzed soil parameters.

**Figure 5 ijerph-20-00455-f005:**
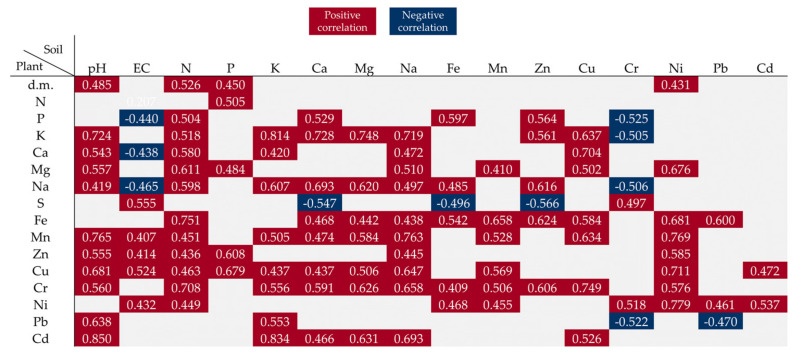
Pearson correlation coefficient showing the influence of soil properties on the features of spring rape and the content of elements in seeds (significance at *p* < 0.05).

**Figure 6 ijerph-20-00455-f006:**
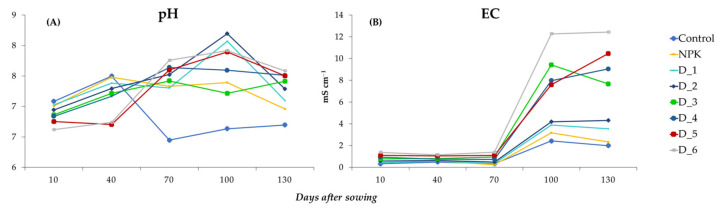
Dynamics of changes in pH (**A**) and EC (**B**) of soil solutions depending on the fertilization applied during the experiment.

**Figure 7 ijerph-20-00455-f007:**
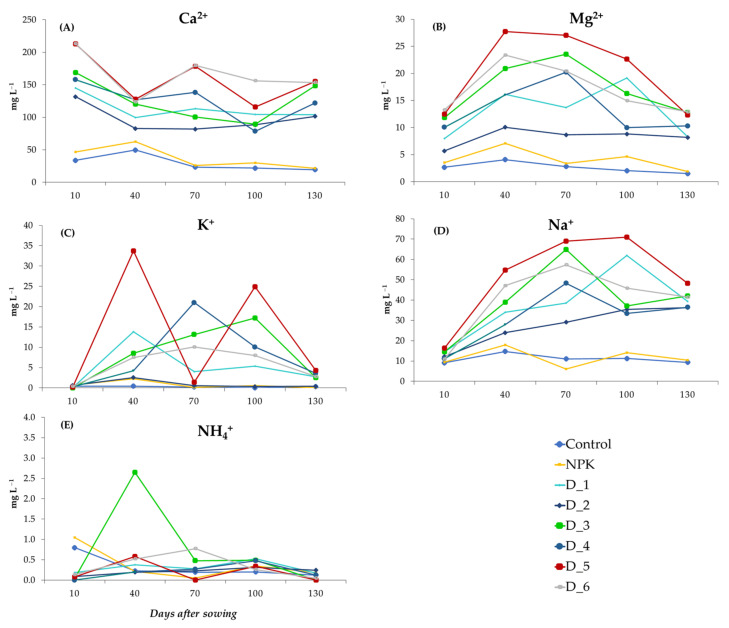
Dynamics of changes in the content of individual inorganic cations: Ca^2+^ (**A**), Mg^2+^ (**B**), K^+^ (**C**), Na^+^ (**D**) and NH_4_^+^ (**E**) contained in the soil solution during the experiment depending on the applied fertilization.

**Figure 8 ijerph-20-00455-f008:**
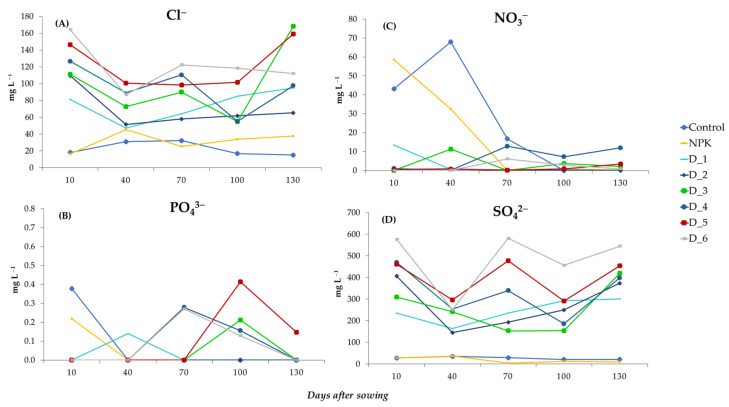
Dynamics of changes in the content of individual anions: Cl^−^ (**A**), PO_4_^3−^ (**B**), NO_3_^−^ (**C**) and SO_4_^2−^ (**D**) contained in the soil solution during the experiment depending on the doses of biomass ash fertilization.

**Figure 9 ijerph-20-00455-f009:**
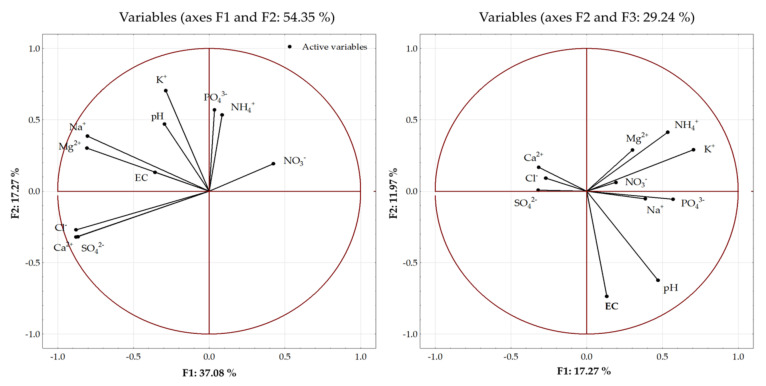
Principal component analysis (PCA) biplot of the distribution of the analyzed parameters of soil solutions.

**Figure 10 ijerph-20-00455-f010:**
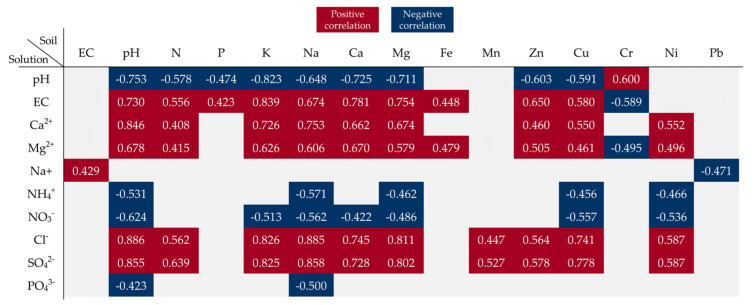
Pearson correlation coefficient map showing the influence of soil parameters on the properties of soil solutions during the experiment (significance at *p* < 0.05).

**Table 1 ijerph-20-00455-t001:** Selected properties of soil used in the pot experiments (mean ± SE).

pH	SOC	Nt	Available Forms of Nutrients *				Total Concentrations of Elements				
			P	K	Mg	Zn	Cu	Ni	Cr	Pb	Cd
H_2_O	%		mg kg^−1^								
6.78 ± 0.12	1.23 ± 0.16	0.09 ± 0.0	7.92 ± 0.7	26.4 ± 2.2	92.1 ± 6.8	43.2 ± 8.6	5.67 ± 0.9	12.2 ± 1.04	31.2 ± 5.01	16.4 ± 1.2	0.34 ± 0.1

Explanation: SOC—soil organic carbon; Nt–total nitrogen; * determination of available forms of nutrients with the Mehlich 3 method [19].

**Table 2 ijerph-20-00455-t002:** Physicochemical properties of ash from biomass combustion used in the experiment (mean ± SE).

Parameter	Units		Total Concentration	Soluble in H_2_O	Soluble in 1 M HCl
pH	-	12.83 ± 0.37	-	-	-
EC	mS cm^−1^	8.81 ± 0.24	-	-	-
C	%	-	2.99 ± 0.21	-	-
N	mg kg^−1^	-	10 ± 0.52	-	-
S		-	4700 ± 50	-	-
P		-	9244 ± 65	-	-
K		-	165,617 ± 123	-	-
Mg		-	13,512 ± 78	-	-
Ca		-	145,081 ± 137	-	-
Na		-	1452 ± 43	-	-
Fe		-	4351 ± 16	0.06 ± 0.01	1239 ± 54
Mn		-	1490 ± 21	0.08 ± 0.01	635 ± 27
Zn		-	423 ± 11	0.02 ± 0.00	395 ± 19
Cu		-	536 ± 28	0.07 ± 0.01	157 ± 16
Ni		-	20.3 ± 1.1	0.01 ± 0.00	15.9 ± 1.8
Cr		-	48.4 ± 2.9	0.09 ± 0.02	32.8 ± 2.7
Pb		-	130 ± 3.7	0.01 ± 0.00	33.8 ± 1.6
Cd		-	2.68 ± 0.81	0.01 ± 0.00	2.49 ± 0.9

**Table 3 ijerph-20-00455-t003:** Content of macroelements in spring rape seeds depending on the fertilization variant (mean ± SE).

Variant	N	P	K	Mg	Ca	Na	S
				g kg^−1^			
Control	30.0 ^a,b^ ± 0.4	4.39 ^b^ ± 0.12	8.65 ^a^ ± 0.54	1.20 ^a^ ± 0.16	4.21 ^a,b^ ± 0.30	0.13 ^a^ ± 0.00	1.77 ^a^ ± 0.06
NPK	↑31.2 ^b^ ± 0.4	↑6.33 ^c^ ± 0.07	↓8.15 ^a^ ± 0.38	↑2.19 ^b^ ± 0.25	↑4.35 ^b^ ± 0.38	↑0.18 ^d^ ± 0.01	↑2.16 ^b^ ± 0.06
D1	↑31.2 ^b^ ± 0.4	↓3.78 ^a^ ± 0.36	↓8.22 ^a^ ± 0.17	↑2.65 ^c,d^ ± 0.37	↓4.13 ^a^ ± 0.33	↓0.12 ^a^ ± 0.00	↑2.32 ^b^ ± 0.05
D2	↑30.3 ^a,b^ ± 0.6	↓3.93 ^a^ ± 0.34	↑10.02 ^b^ ± 0.54	↑2.79 ^d^ ± 0.20	↑4.89 ^c^ ± 0.19	↑0.15 ^b,c^ ± 0.00	↑2.15 ^b^ ± 0.04
D3	↓29.0 ^a^ ± 0.5	↑6.10 ^c^ ± 0.20	↑10.36 ^c^ ± 0.36	↑3.00 ^e^ ± 0.58	↑5.09 ^d^ ± 0.22	↑0.16 ^c,d^ ± 0.00	↑1.82 ^a^ ± 0.08
D4	↑30.6 ^a,b^ ± 0.4	↑7.72 ^e^ ± 0.35	↑11.12 ^d^ ± 0.54	↑3.01 ^e^ ± 0.35	↑4.84 ^c^ ± 0.22	↑0.22 ^e^ ± 0.00	↓1.55 ^a^ ± 0.08
D5	↑31.2 ^b^ ± 0.2	↑7.08 ^d^ ± 0.26	↑12.14 ^e^ ± 0.43	↑2.57 ^b,c^ ± 0.33	↑4.77 ^c^ ± 0.34	↑0.22 ^e^ ± 0.00	↓1.73 ^a^ ± 0.06
D6	↑31.5 ^b^ ± 0.4	↑6.16 ^c^ ± 0.27	↑12.09 ^e^ ± 0.49	↑2.62 ^b,c^ ± 0.17	↑4.80 ^c^ ± 0.18	↑0.21 ^e^ ± 0.00	↓1.75 ^a^ ± 0.04

The values are based on dry weight. Mean values ± standard error. (a–e) Identical superscripts denote no significant (*p* < 0.05) differences between the experimental objects according to the post-hoc Tukey HSD test. The red/blue arrows indicate higher/lower values compared to the control, respectively.

**Table 4 ijerph-20-00455-t004:** Content of trace elements in spring rape seeds depending on the fertilization variant (mean ± SE).

Variant	Fe	Mn	Zn	Cu	Ni	Cr	Pb	Cd
	mg kg^−1^							
Control	22.6 ^a^ ± 0.3	6.1 ^a^ ± 0.1	13.7 ^a^ ± 0.4	1.84 ^a^ ± 0.06	0.51 ^a,b^ ± 0.01	0.20 ^a^ ± 0.00	0.25 ^b,c^ ± 0.02	0.06 ^b^ ± 0.00
NPK	↑27.8 ^a^ ± 0.2	↑7.9 ^a^ ± 0.1	↑27.9 ^b,c^ ± 0.4	↑2.51 ^b^ ± 0.08	↓0.50 ^a,b^ ± 0.01	↑0.21 ^a^ ± 0.01	↓0.12 ^a^ ± 0.01	↓0.04 ^a^ ± 0.00
D1	↑23.3 ^a^ ± 0.2	↑27.6 ^c^ ± 0.1	↑30.7 ^e^ ± 0.3	↑2.97 ^c^ ± 0.07	↑1.08 ^a^ ± 0.03	0.20 ^a^ ± 0.01	↓0.22 ^b^ ± 0.01	↑0.07 ^b^ ± 0.00
D2	↑46.5 ^c^ ± 0.4	↑27.9 ^c^ ± 0.2	↑30.1 ^d,e^ ± 0.1	↑2.69 ^b^ ± 0.02	↑0.88 ^c^ ± 0.06	↑2.55 ^b^ ± 0.04	↑0.30 ^c,d^ ± 0.01	↑0.09 ^c^ ± 0.00
D3	↑48.5 ^c^ ± 0.5	↑27.9 ^c^ ± 0.4	↑28.9 ^c,d^ ± 0.3	↑2.46 ^b^ ± 0.05	↑1.05 ^d^ ± 0.04	↑2.87 ^d^ ± 0.04	↑0.33 ^d,e^ ± 0.01	↑0.07 ^b^ ± 0.00
D4	↑82.1 ^d^ ± 1.7	↑28.3 ^c^ ± 0.2	↑28.3 ^c^ ± 0.2	↑2.78 ^b,c^ ± 0.03	↑1.21 ^e^ ± 0.01	↑3.87 ^e^ ± 0.08	↓0.13 ^a^ ± 0.01	0.06 ^b^ ± 0.00
D5	↑39.8 ^b,c^ ± 1.0	↑25.4 ^b^ ± 0.4	↑27.0 ^b^ ± 0.2	↑2.73 ^b,c^ ± 0.02	↓0.43 ^a^ ± 0.03	↑2.75 ^c,d^ ± 0.04	↑0.39 ^f^ ± 0.01	↑0.10 ^c^ ± 0.00
D6	↑35.4 ^b^ ± 1.9	↑24.8 ^b^ ± 0.9	↑26.9 ^b^ ± 0.1	↑2.79 ^b,c^ ± 0.03	↑0.61 ^b^ ± 0.03	↑2.56 ^b,c^ ± 0.05	↑0.36 ^e,f^ ± 0.01	↑0.11 ^c^ ± 0.00

The values are based on dry weight. Mean values ± standard error. (a–f) Identical superscripts denote no significant (*p* < 0.05) differences between the experimental objects according to the post-hoc Tukey HSD test. The red/blue arrows indicate higher/lower values compared to the control, respectively.

**Table 5 ijerph-20-00455-t005:** Effect of ash from biomass combustion on selected properties of soil (mean ± SE).

Variant	pH	EC	Total Macronutrients (mg kg^−1^)					
	[H_2_O]	[mS cm^−1^]	N	P	K	Mg	Ca	Na
Control	6.48 ^a^ ± 0.06	0.10 ^a^ ± 0.00	1927 ^a^ ± 22	979 ^a^ ± 27	1196 ^b^ ± 19	1294 ^a,b^ ± 4	1558 ^a,b^ ± 21	138 ^a^ ± 2
NPK	↑6.59 ^a^ ± 0.04	↑0.13 ^a^ ± 0.00	↑2058 ^b,c^ ± 20	↑1891 ^c^ ± 17	↓1110 ^a^ ± 21	↓1247 ^a^ ± 17	↓1481 ^a^ ± 21	↓136 ^a^ ± 4
D1	↑6.92 ^b^ ± 0.08	↑0.49 ^e^ ± 0.01	↑1991 ^a,b^ ± 11	↑1845 ^c^ ± 19	↑1237 ^b^ ± 22	↑1350 ^b,c^ ± 20	↑1639 ^c,d^ ± 13	↑157 ^b,c^ ± 3
D2	↑7.09 ^d^ ± 0.02	↑0.16 ^b,c^ ± 0.00	↑2122 ^c^ ± 18	↑1725 ^d^ ± 23	↑1321 ^c^ ± 14	↑1364 ^b,c^ ± 14	↓1521 ^a,b^ ± 15	↑164 ^c^ ± 4
D3	↑6.86 ^c^ ± 0.07	↑0.17 ^c,d^ ± 0.00	↑2081 ^c^ ± 17	↑1398 ^b^ ± 19	↑1195 ^b^ ± 11	1294 ^a,b^ ± 12	↑1581 ^b,c^ ± 23	↑147 ^a,b^ ± 3
D4	↑6.83 ^c^ ± 0.06	↑0.16 ^b,c^ ± 0.00	↑2302 ^d^ ± 13	↑1481 ^b^ ± 19	↑1337 ^c^ ± 15	↑1448 ^d,e^ ± 20	↑1842 ^e^ ± 22	↑166 ^c^ ± 3
D5	↑7.09 ^d^ ± 0.02	↑0.19 ^d^ ± 0.00	↑1988 ^a,b^ ± 14	↑1489 ^b^ ± 10	↑1385 ^c^ ± 13	↑1391 ^c,d^ ± 15	↑1711 ^d^ ± 14	↑164 ^c^ ± 4
D6	↑7.29 ^e^ ± 0.04	↑0.13 ^a^ ± 0.00	↑2230 ^d^ ± 12	↑2326 ^e^ ± 33	↑1580 ^d^ ± 18	↑1491 ^e^ ± 14	↑1865 ^e^ ± 28	↑171 ^c^ ± 4

The values are based on dry weight. Mean values ± standard error. (a–e) Identical superscripts denote no significant (*p* < 0.05) differences between the experimental objects according to the post-hoc Tukey HSD test. The red/blue arrows indicate higher/lower values compared to the control, respectively.

**Table 6 ijerph-20-00455-t006:** Effect of ash from biomass combustion on the content of trace elements in soil (mean ± SE).

Variants	Fe	Mn	Zn	Cu	Ni	Cr	Pb	Cd
					mg kg^−1^			
Control	2421 ^b^ ± 31	127 ^a^ ± 1	41.1 ^a,b^ ± 0.7	4.71 ^a^ ± 0.16	14.6 ^a^ ± 0.3	29.3 ^b,c^ ± 0.5	15.6 ^c^ ± 0.1	0.25 ^a^ ± 0.01
NPK	↑2436 ^b^ ± 24	↑129 ^a^ ± 2	↓40.7 ^a^ ± 0.3	↓4.54 ^a^ ± 0.25	↑14.9 ^a^ ± 0.1	↓28.7 ^b,c^ ± 0.3	↓15.3 ^b,c^ ± 0.1	0.25 ^a^ ± 0.01
D1	↑2870 ^d^ ± 19	↑163 ^c^ ± 2	↑42.0 ^a,b^ ± 0.4	↑4.88 ^a^ ± 0.37	↑17.8 ^c^ ± 0.1	↑30.7 ^c,d^ ± 0.4	↓14.9 ^a,b^ ± 0.2	↑0.32 ^b,c^ ± 0.01
D2	↓1983 ^a^ ± 9	↑157 ^b,c^ ± 6	↓40.4 ^a^ ± 0.4	↑6.14 ^c^ ± 0.20	↑16.3 ^b^ ± 0.2	↑31.9 ^d^ ± 0.3	↓15.2 ^b,c^ ± 0.1	↓0.22 ^a^ ± 0.01
D3	↑2701 ^c^ ± 11	↓120 ^a^ ± 1	↑43.1 ^b^ ± 0.3	↑5.43 ^b^ ± 0.58	↑16.4 ^b^ ± 0.6	↓27.4 ^a,b^ ± 0.6	↓15.5 ^b,c^ ± 0.2	0.25 ^a^ ± 0.01
D4	↑3481 ^e^ ± 15	↑204 ^d^ ± 4	↑49.5 ^c^ ± 0.5	↑5.96 ^c^ ± 0.35	↑19.0 ^d^ ± 0.2	↑30.4 ^c,d^ ± 0.5	↑16.7 ^d^ ± 0.1	↑0.34 ^c^ ± 0.01
D5	↑2630 ^c^ ± 26	↑164 ^c^ ± 3	↑41.4 ^a,b^ ± 0.5	↑4.99 ^a,b^ ± 0.33	↑16.3 ^b^ ± 0.3	↓26.3 ^a^ ± 0.4	↓14.3 ^a^ ± 0.2	↓0.24 ^a^ ± 0.01
D6	↑2916 ^d^ ± 28	↑144 ^b^ ± 3	↑48.2 ^c^ ± 0.3	↑6.27 ^c^ ± 0.17	↑16.5 ^b^ ± 0.4	↓26.6 ^a^ ± 0.3	↑15.7 ^c^ ± 0.1	↑0.30 ^b^ ± 0.01

The values are based on dry weight. Mean values ± standard error. (a–e) Identical superscripts denote no significant (*p* < 0.05) differences between the experimental objects according to the post-hoc Tukey HSD test. The red/blue arrows indicate higher/lower values compared to the control, respectively.

## Data Availability

The entire set of raw data presented in this study is available on request from the corresponding author.

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
