# Peer review of "Short-Term Effect of Fly Ash from Biomass Combustion on Spring Rape Plants Growth, Nutrient, and Trace Elements Accumulation, and Soil Properties"

_ijerph, 2022, doi:10.3390/ijerph20010455_

Round 1
Reviewer 1 Report
Langauge correction to be made.
Certain Concepts highlighted in the materials and methods section were not projected in introduction and conclusion, eg.,PCA.
The discussion can be made clear to improve the readability.

Author Response
The authors are grateful for the contribution of the reviewer process. As a result, numerous errors were removed, and various manuscript sections were improved. Please find in the attachment our response.

Reviewer 2 Report
The use of fly ashes to improve soil properties is becoming quite common due to the positive properties of fly ashes. On the other hand, we have a problem with the management of fly ashes, so using them as soil additives is fully justified.
Summary and introduction well written. The material and methods chapter is written exhaustively and in great detail.
The research results have been presented in a very broad way, in addition, numerous graphics and charts have been presented, which are clear and facilitate the interpretation of the obtained research results. The test results are very meticulously described, which greatly facilitates their reception. Sufficient discussion and conclusions. I accept for printing in its current form.
Author Response
The authors are grateful for the contribution of the reviewer process and the positive evaluation of the manuscript.

Author Response

(The authors gave the same response as above.)

Reviewer 4 Report
This study have assessed the short-term impact of BA fertilization on the properties of soils and soil solutions. It is interesting; however, the authors are advised to attend the following observations.
- It is suggested to simplify the abstract of this study.
- It is difficult for the reader to find the important experimental records in the table 3 to table 6. So, it is suggested to reorganize the experimental records and exhibited by Figures to show the special records in the experiment.
- In line 131, 300 μE m−2 s−1 should be 300 μE m−2 s−1 .
Author Response

(The authors gave the same response as above.)

Round 2
Reviewer 3 Report
I think the current version is acceptable.
Reviewer 4 Report
The manuscript has been improved greatly.